# MEMORIZATION PRECEDES GENERATION: LEARNING UNSUPERVISED GANS WITH MEMORY NETWORKS

**Youngjin Kim, Minjung Kim & Gunhee Kim**
Department of Computer Science and Engineering, Seoul National University, Seoul, Korea
{youngjin.kim,minjung.kim,gunhee.kim}@vision.snu.ac.kr

## ABSTRACT

We propose an approach to address two issues that commonly occur during training of unsupervised GANs. First, since GANs use only a continuous latent distribution to embed multiple classes or clusters of data, they often do not correctly handle the structural discontinuity between disparate classes in a latent space. Second, discriminators of GANs easily forget about past generated samples by generators, incurring instability during adversarial training. We argue that these two infamous problems of unsupervised GAN training can be largely alleviated by a learnable *memory network* to which both generators and discriminators can access. Generators can effectively learn representation of training samples to understand underlying cluster distributions of data, which ease the structure discontinuity problem. At the same time, discriminators can better memorize clusters of previously generated samples, which mitigate the forgetting problem. We propose a novel end-to-end GAN model named *memoryGAN*, which involves a memory network that is unsupervisedly trainable and integrable to many existing GAN models. With evaluations on multiple datasets such as Fashion-MNIST, CelebA, CIFAR10, and Chairs, we show that our model is probabilistically interpretable, and generates realistic image samples of high visual fidelity. The memoryGAN also achieves the state-of-the-art inception scores over unsupervised GAN models on the CIFAR10 dataset, without any optimization tricks and weaker divergences.

## 1 INTRODUCTION

Generative Adversarial Networks (GANs) (Goodfellow et al., 2014) are one of emerging branches of unsupervised models for deep neural networks. They consist of two neural networks named *generator* and *discriminator* that compete each other in a zero-sum game framework. GANs have been successfully applied to multiple generation tasks, including image syntheses (*e.g.* (Reed et al., 2016b; Radford et al., 2016; Zhang et al., 2016a)), image super-resolution (*e.g.* (Ledig et al., 2017; Snderby et al., 2017)), image colorization (*e.g.* (Zhang et al., 2016b)), to name a few. Despite such remarkable progress, GANs are notoriously difficult to train. Currently, such training instability problems have mostly tackled by finding better distance measures (*e.g.* (Li et al., 2015; Nowozin et al., 2016; Arjovsky & Bottou, 2017; Arjovsky et al., 2017; Gulrajani et al., 2017; Warde-Farley & Bengio, 2017; Mroueh et al., 2017; Mroueh & Sercu, 2017)) or regularizers (*e.g.* (Salimans et al., 2016; Metz et al., 2017; Che et al., 2017; Berthelot et al., 2017)).

We aim at alleviating two undesired properties of unsupervised GANs that cause instability during training. The first one is that GANs use a unimodal continuous latent space (*e.g.* Gaussian distribution), and thus fail to handle structural discontinuity between different classes or clusters. It partly attributes to the infamous mode collapsing problem. For example, GANs embed both *building* and *cats* into a common continuous latent distribution, even though there is no intermediate structure between them. Hence, even a perfect generator would produce unrealistic images for some latent codes that reside in transitional regions of two disparate classes. Fig.1 (a,c) visualize this problem with examples of affine-transformed MNIST and Fashion-MNIST datasets. There always exist latent regions that cause unrealistic samples (red boxes) between different classes (blue boxes at the corners). Another problem is the discriminator's forgetting behavior about past synthesized samples by the generator, during adversarial training of GANs. The catastrophic forgetting has explored in deep network research, such as (Kirkpatrick et al., 2016; Kemker et al., 2017); in the context of

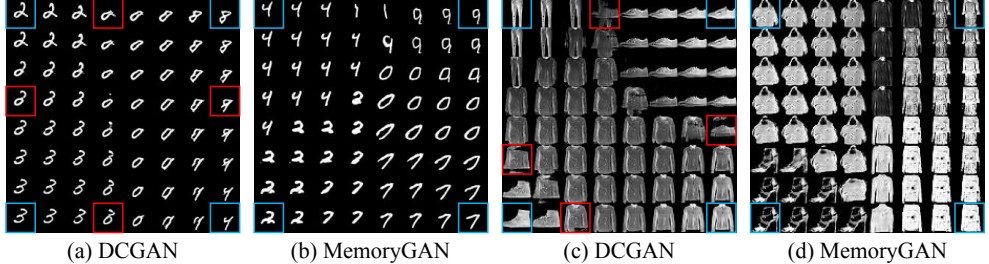

| (a) DCGAN | (b) MemoryGAN | (c) DCGAN | (d) MemoryGAN |

Figure 1: Evidential visualization of structural discontinuity in a latent space with examples of affine-MNIST in (a)–(b) and Fashion-MNIST in (c)–(d). In each set, we first generate four images at the four corners in blue boxes by randomly sampling four latent variables ($z$'s for DCGAN and $(z, c)$'s for our memoryGAN). We then generate 64 images by interpolating latent variables between them. In memoryGAN, $c$ is a memory index, and it is meaningless to directly interpolate over $c$. Therefore, for the four sampled $c$'s, we take their key values $K_c$ and interpolate over $K_c$ in the query space. For each interpolated vector $\hat{K}_c$, we find the memory slot $c = \arg\max_c p(c|\hat{K}_c)$, *i.e.* the memory index whose posterior is the highest with respect to $\hat{K}_c$. (a,c) In unsupervised DCGAN, unrealistic samples (depicted in red boxes) always exist in some interpolated regions, because different classes are embedded in a unimodal continuous latent space. (b,d) In our memoryGAN, samples gradually change, but no structural discontinuity occurs.

GANs, Shrivastava et al. (2017) address the problem that the discriminator often focuses only on latest input images. Since the loss functions of the two networks depend on each other's performance, such forgetting behavior results in serious instability, including causing the divergence of adversarial training, or making the generator re-introduce artifacts that the discriminator has forgotten.

We claim that a simple memorization module can effectively mitigate both instability issues of unsupervised GAN training. First, to ease the structure discontinuity problem, the memory can learn representation of training samples that help the generator better understand the underlying class or cluster distribution of data. Fig.1 (b,d) illustrate some examples that our memory network successfully learn implicit image clusters as key values using a Von Mises-Fisher (vMF) mixture model. Thus, we can separate the modeling of discrete clusters from the embedding of data attributes (*e.g.* styles or affine transformation in images) on a continuous latent space, which can ease the structural discontinuity issue. Second, the memory network can mitigate the forgetting problem by learning to memorize clusters of previously generated samples by the generator, including even rare ones. It makes the discriminator trained robust to temporal proximity of specific batch inputs.

Based on these intuitions, we propose a novel end-to-end GAN model named *memoryGAN*, that involves a life-long memory network to which both generator and discriminator can access. It can learn multi-modal latent distributions of data in an unsupervised way, without any optimization tricks and weaker divergences. Moreover, our memory structure is orthogonal to the generator and discriminator design, and thus integrable with many existing variants of GANs.

We summarize the contributions of this paper as follows.

- We propose *memoryGAN* as a novel unsupervised framework to resolve the two key instability issues of existing GAN training, the structural discontinuity in a latent space and the forgetting problem of GANs. To the best of our knowledge, our model is a first attempt to incorporate a memory network module with unsupervised GAN models.

- In our experiments, we show that our model is probabilistically interpretable by visualizing data likelihoods, learned categorical priors, and posterior distributions of memory slots. We qualitatively show that our model can generate realistic image samples of high visual fidelity on several benchmark datasets, including Fashion-MNIST (Xiao et al., 2017), CelebA (Liu et al., 2015), CIFAR10 (Krizhevsky, 2009), and Chairs (Aubry et al., 2014). The memoryGAN also achieves the state-of-the-art inception scores among unsupervised GAN models on the CIFAR10 dataset. The code is available at `https://github.com/whyjay/memoryGAN`.

## 2 RELATED WORK

**Memory networks**. Augmenting neural networks with memory has been studied much (Bahdanau et al., 2014; Graves et al., 2014; Sukhbaatar et al., 2015). In these early memory models, computational requirement necessitates the memory size to be small. Some networks such as (Weston et al., 2014; Xu et al., 2016) use large-scale memory but its size is fixed prior to training. Recently, Kaiser et al. (2017) extend (Santoro et al., 2016; Rae et al., 2016) and propose a large-scale life-long memory network, which does not need to be reset during training. It exploits nearest-neighbor search for efficient memory lookup, and thus scales to a large memory size. Unlike previous approaches, our memory network in the discriminator is designed based on a mixture model of Von Mises-Fisher distributions (vMF) and an incremental EM algorithm. We will further specify the uniqueness of read, write, and sampling mechanism of our memory network in section 3.1.

There have been a few models that strengthen the memorization capability of generative models. Li et al. (2016) use additional trainable parameter matrices as a form of memory for deep generative models, but they do not consider the GAN framework. Arici & Celikyilmaz (2016) use RBMs as an associative memory, to transfer some knowledge of the discriminator's feature distribution to the generator. However, they do not address the two problems of our interest, the structural discontinuity and the forgetting problem; as a result, their memory structure is far different from ours.

**Structural discontinuity in a latent space**. Some previous works have addressed this problem by splitting the latent vector into multiple subsets, and allocating a separate distribution to each data cluster. Many conditional GAN models concatenate random noises with vectorized external information like class labels or text embedding, which serve as cluster identifiers, (Mirza & Osindero, 2014; Gauthier, 2015; Reed et al., 2016b; Zhang et al., 2016a; Reed et al., 2016a; Odena et al., 2017; Dash et al., 2017). However, such networks are not applicable to an unsupervised setting, because they require supervision of conditional information. Some image editing GANs and cross-domain transfer GANs extract the content information from input data using auxiliary encoder networks (Yan et al., 2016; Perarnau et al., 2016; Antipov et al., 2017; Denton et al., 2016; Lu et al., 2017; Zhang et al., 2017; Isola et al., 2017; Taigman et al., 2017; Kim et al., 2017; Zhu et al., 2017). However, their auxiliary encoder networks transform only given images rather than entire sample spaces of generation. On the other hand, memoryGAN learns the cluster distributions of data without supervision, additional encoder networks, and source images to edit from.

InfoGAN (Chen et al., 2016) is an important unsupervised GAN framework, although there are two key differences from our memoryGAN. First, InfoGAN implicitly learns the latent cluster information of data into small-sized model parameters, while memoryGAN explicitly maintains the information on a learnable life-long memory network. Thus, memoryGAN can easily keep track of cluster information stably and flexibly without suffering from forgetting old samples. Second, memoryGAN supports various distributions to represent priors, conditional likelihoods, and marginal likelihoods, unlike InfoGAN. Such interpretability is useful for designing and training the models.

**Fogetting problem**. It is a well-known problem that the discriminator of GANs is prone to forgetting past samples that the generator synthetizes. Multiple studies such as (Salimans et al., 2016; Shrivastava et al., 2017) address the forgetting problem in GANs and make a consensus on the need for memorization mechanism. In order for GANs to be less sensitive to temporal proximity of specific batch inputs, Salimans et al. (2016) add a regularization term of the $\ell_2$-distance between current network parameters and the running average of previous parameters, to prevent the network from revisiting previous parameters during training. Shrivastava et al. (2017) modify the adversarial training algorithm to involve a buffer of synthetic images generated by the previous generator. Instead of adding regularization terms or modifying GAN algorithms, we explicitly increase the model's memorization capacity by introducing a life-long memory into the discriminator.

## 3 THE MEMORYGAN

Fig.2 shows the proposed *memoryGAN* architecture, which consists of a novel discriminator network named *Discriminative Memory Network* (DMN) and a generator network as *Memory Conditional Generative Network* (MCGN). We describe DMN and MCGN in section 3.1–3.3, and then discuss how our memoryGAN can resolve the two instability issues in section 3.4.

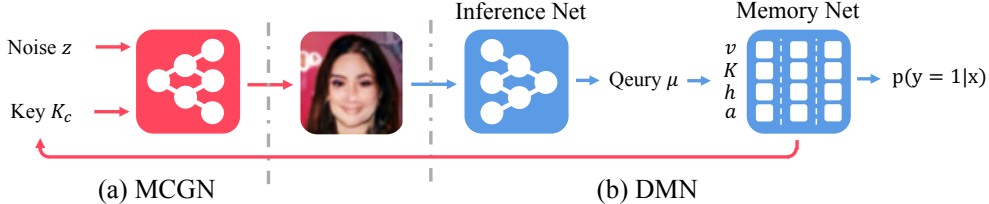

Figure 2: The memoryGAN architecture. (a) The memory conditional generative network (MCGN) samples a memory slot index $c$ from a categorical prior distribution over memory slots, and use the mean direction of corresponding mixture component $K_c$ as conditional cluster information. (b) The discriminative memory network (DMN) consists of an inference network and a memory module.

### 3.1 THE DISCRIMINATIVE MEMORY NETWORK

The DMN consists of an inference network and a memory network. The inference network $\mu$ is a convolutional neural network (CNN), whose input is a datapoint $x \in \mathbb{R}^D$ and output is a normalized query vector $q = \mu(x) \in \mathbb{R}^M$ with $\|q\| = 1$. Then the memory module takes the query vector as input and decides whether $x$ is real or fake: $y \in \{0, 1\}$.

We denote the memory network by four tuples: $\mathcal{M} = (K, v, a, h)$. $K \in \mathbb{R}^{N \times M}$ is a memory key matrix, where $N$ is the memory size (*i.e.* the number of memory slots) and $M$ is the dimension. $v \in \{0, 1\}^N$ is a memory value vector. Conceptually, each key vector stores a cluster center representation learned via the vMF mixture model, and its corresponding key value is a target indicating whether the cluster center is real or fake. $a \in \mathbb{R}^N$ is a vector that tracks the age of items stored in each memory slot, and $h \in \mathbb{R}^N$ is the slot histogram, where each $h_i$ can be interpreted as the effective number of data points that belong to $i$-th memory slot. We initialize them as follows: $K = 0$, $a = 0$, $h = 10^{-5}$, and $v$ as a flat categorical distribution.

Our memory network partly borrows some mechanisms of the *life-long memory network* (Kaiser et al., 2017), that is free to increase its memory size, and has no need to be reset during training. Our memory also uses the $k$-nearest neighbor indexing for efficient memory lookup, and adopts the least recently used (LRU) scheme for memory update. Nonetheless, there are several novel features of our memory structure as follows. First, our method is probabilistically interpretable; we can easily compute the data likelihood, categorical prior, and posterior distribution of memory indices. Second, our memory learns the approximate distribution of a query by maximizing likelihood using an incremental EM algorithm (see section 3.1.2). Third, our memory is optimized from the GAN loss, instead of the memory loss as in (Kaiser et al., 2017). Fifth, our method tracks the slot histogram to determine the degree of contributions of each sample to the slot.

Below we first discuss how to compute the discriminative probability $p(y|x)$ for a given real or fake sample $x$ at inference, using the memory and the learned inference network (section 3.1.1). We then explain how to update the memory during training (section 3.1.2).

#### 3.1.1 THE DISCRIMINATIVE OUTPUT

For a given sample $x$, we first find out which memory slots should be referred for computing the discriminative probability. We use $c \in \{1, 2, \ldots, N\}$ to denote a memory slot index. We represent the posterior distribution over memory indices using a Von Mises-Fisher (vMF) mixture model as

$$p(c = i|x) = \frac{p(x|c = i)p(c = i)}{\sum_{j=1}^{N} p(x|c = j)p(c = j)} = \frac{\exp(\kappa K_i^T \mu(x))p(c = i)}{\sum_{j=1}^{N} \exp(\kappa K_j^T \mu(x))p(c = j)} \quad (1)$$

where the likelihood $p(x|c = i) = C(\kappa) \exp(\kappa K_i^T \mu(x))$ with a constant concentration parameter $\kappa = 1$. Remind that vMF is, in effect, equivalent to a properly normalized Gaussian distribution defined on a unit sphere. The categorical prior of memory indices, $p(c)$, is obtained by nomalizing the slot histogram: $p(c = i) = \frac{h_i + \beta}{\sum_{j=1}^{N}(h_j + \beta)}$, where $\beta(= 10^{-8})$ is a small smoothing constant for numerical stability. Using $p(y = 1|c = i, x) = v_i$ (*i.e.* the key value of memory slot $i$), we can estimate the discriminative probability $p(y = 1|x)$ by marginalizing the joint probability

$p(y = 1, c|x)$ over $c$:

$$p(y = 1|x) = \sum_{i=1}^{N} p(y = 1|c = i, x)p(c = i|x) = \sum_{i=1}^{N} v_i p(c = i|x) = \mathbb{E}_{i \sim p(c|x)}[v_i]. \quad (2)$$

However, in practice, it is not scalable to exhaustively sum over the whole memory of size $N$ for every sample $x$; we approximate Eq.(2) considering only top-$k$ slots $S = \{s_1, ..., s_k\}$ with the largest posterior probabilities (*e.g.* we use $k = 128 \sim 256$):

$$S = \underset{c_1,...,c_k}{\operatorname{argmax}} p(c|x) = \underset{c_1,...,c_k}{\operatorname{argmax}} p(x|c)p(c) = \underset{c_1,...,c_k}{\operatorname{argmax}} \exp(\kappa K_c^T \mu(x))(h_c + \beta) \quad (3)$$

where $p(x|c)$ is vMF likelihood and $p(c)$ is prior distribution over memory indicies. We omit the normalizing constant of the vMF likelihood and the prior denominator since they are constant over all memory slots. Once we obtain $S$, we approximate the discriminative output as

$$p(y|x) \approx \frac{\sum_{i \in S} v_i p(x|c = i)p(c = i)}{\sum_{j \in S} p(x|c = j)p(c = j)}. \quad (4)$$

We clip $p(y|x)$ into $[\epsilon, 1 - \epsilon]$ with a small constant $\epsilon = 0.001$ for numerical stability.

### 3.1.2 MEMORY UPDATE

Memory keys and values are updated during the training. We adopt both a conventional memory update mechanism and an incremental EM algorithm. We denote a training sample $x$ and its label $y$ that is 1 for real and 0 for fake. For each $x$, we first find $k$-nearest slots $S_y$ as done in Eq.(3), except using the conditional posterior $p(c|x, v_c = y)$ instead of $p(c|x)$. This change is required to consider only the slots that belong to the same class with $y$ during the following EM algorithm. Next we update the memory in two different ways, according to whether $S_y$ contains the correct label $y$ or not. If there is no correct label in the slots of $S_y$, we find the oldest memory slot by $n_a = \operatorname{argmax}_{i \in \{1,...,N\}} a_i$, and copy the information of $x$ on it: $K_{n_a} \leftarrow q = \mu(x)$, $v_{n_a} \leftarrow y$, $a_{n_a} \leftarrow 0$, and $h_{n_a} \leftarrow \frac{1}{N} \sum_{i=1}^{N} h_i$. If $S$ contains the correct label, memory keys are updated to partly include the information of new sample $x$, via the following modified incremental EM algorithm for $T$ iterations. In the expectation step, we compute posterior $\gamma_i^t = p(c_i|x)$ for $i \in S_y$, by applying previous keys $\hat{K}_i^{t-1}$ and $\hat{h}_i^{t-1}$ to Eq.(1). In the maximization step, we update the required sufficient statistics as

$$\hat{h}_i^t \leftarrow \hat{h}_i^{t-1} + \gamma^t - \gamma^{t-1}, \quad \hat{K}_i^t \leftarrow \hat{K}_i^{t-1} + \frac{\gamma^t - \gamma^{t-1}}{\hat{h}_i^t}(q_i - \hat{K}_i^t) \quad (5)$$

where $t \in 1, ..., T$, $\gamma^0 = 0$, $\hat{K}_i^1 = K_i$, $\hat{h}_i^1 = \alpha h_i$ and $\alpha = 0.5$. After $T$ iterations, we update the slots of $S_y$ by $K_i \leftarrow \hat{K}_i^t$ and $h_i \leftarrow \hat{h}_i^t$. The decay rate $\alpha$ controls how much it exponentially reduces the contribution of old queries to the slot position of the mean directions of mixture components. $\alpha$ is often critical for the performance, give than old queries that were used to update keys are unlikely to be fit to the current mixture distribution, since the inference network $\mu$ itself updates during the training too. Finally, it is worth noting that this memory updating mechanism is orthogonal to any adversarial training algorithm, because it is performed separately while the discriminator is updated. Moreover, adding our memory module does not affects the running speed of the model at test time, since the memory is updated only at training time.

### 3.2 THE MEMORY-CONDITIONAL GENERATIVE NETWORK

Our generative network MCGN is based on the conditional generator of InfoGAN (Chen et al., 2016). However, one key difference is that the MCGN synthesizes a sample not only conditioned on a random noise vector $z \in \mathbb{R}^{D_z}$, but also on conditional memory information. That is, in addition to sample $z$ from a Gaussian, the MCGN samples a memory index $i$ from $P(c = i|v_c = 1) = \frac{h_i v_i}{\sum_j^N h_j v_j}$, which reflects the exact appearance frequency of the memory cell $i$ within real data. Finally, the generator synthesizes a fake sample from the concatenated representation $[K_i, z]$, where $K_i$ is a key vector of the memory index $i$.

---

**Algorithm 1** Training algorithm of *memoryGAN*. $\phi$ is parameters of discriminator, $\theta$ is parameters of generator. $\alpha = 0.5, \eta = 2 \cdot 10^{-4}$.

---

1: **for** number of training iterations **do**
2:     Sample a minibatch of examples $x$ from training data.
3:     Sample a minibatch of noises $z \sim p(z)$ and memory indices $c \sim p(c|v = 1)$.
4:     Update by the gradient descent $\phi \leftarrow \phi - \eta \nabla_\phi L_D$
5:     Find $S_y$ for each data in the minibatch
6:     Initialize $\gamma_s^0 \leftarrow 0, \hat{h}_s^0 \leftarrow \alpha h_s$ and $\hat{K}_s^0 \leftarrow K_s$ for $s \in S_y$
7:     **for** number of EM iterations **do**
8:         Estimate posterior $\gamma_s^t$ for $s \in S_y$
9:         $\hat{h}_s^t \leftarrow \hat{h}_s^{t-1} + \gamma_s^t - \gamma_s^{t-1}$ for $s \in S_y$
10:         $\hat{K}_s^t \leftarrow \hat{K}_s^{t-1} + \frac{\gamma_s^t - \gamma_s^{t-1}}{\hat{h}_s^t}(q_i - \hat{K}_s^t)$ for $s \in S_y$
11:     Update vMF mixture model $h_{s_y} \leftarrow \hat{h}_{s_y}^T, K_{s_y} \leftarrow \hat{K}_{s_y}^T$ for $s_y \in S_y$
12:     Sample a minibatch of noises $z \sim p(z)$ and memory indices $c \sim p(c|v = 1)$.
13:     Update by gradient descent $\theta \leftarrow \theta - \eta \nabla_\theta L_G$

---

Unlike other conditional GANs, the MCGN requires neither additional annotation nor any external encoder network. Instead, MCGN makes use of conditional memory information that the DMN learns in an unsupervised way. Recall that the DMN learns the vMF mixture memory with only the query representation $q = \mu(x)$ of each sample $x$ and its indicator $y$.

Finally, we summarize the training algorithm of memoryGAN in Algorithm 1.

### 3.3 THE OBJECTIVE FUNCTION

The objective of memoryGAN is based on that of InfoGAN (Chen et al., 2016), which maximizes the mutual information between a subset of latent variables and observations. We add another mutual information loss term between $K_i$ and $G(z, K_i)$, to ensure that the structural information is consistent between a sampled memory information $K_i$ and a generated sample $G(z, K_i)$ from it:

$$I(K_i; G(z, K_i)) \geq H(K_i) - \hat{I} - \log C(\kappa) \tag{6}$$

where $\hat{I}$ is the expectation of negative cosine similarity $\hat{I} = -E_{x \sim G(z, K_i)}[\kappa K_i^T \mu(x)]$. We defer the derivation of Eq.(6) to Appendix A.1. Finally, the memoryGAN objective can be written with the lower bound of mutual information, with a hyperparameter $\lambda$ (*e.g.* we use $\lambda = 2 \cdot 10^{-6}$) as

$$\mathcal{L}_D = -E_{x \sim p(x)}[\log D(x)] - E_{(z,c) \sim p(z,c)}[\log(1 - D(G(z, K_i)))] + \lambda \hat{I}. \tag{7}$$

$$\mathcal{L}_G = E_{(z,c) \sim p(z,c)}[\log(1 - D(G(z, K_i)))] + \lambda \hat{I}. \tag{8}$$

### 3.4 HOW DOES MEMORYGAN MITIGATE THE TWO INSTABILITY ISSUES?

Our memoryGAN implicitly learns the joint distribution $p(x, z, c) = p(x|z, c)p(z)p(c)$, for which we assume the continuous variable $z$ and the discrete memory variable $c$ are independent. This assumption reflects our intuition that when we synthesize a new sample (*e.g.* an image), we separate the modeling of its class or cluster (*e.g.* an object category) from the representation of other image properties, attributes, or styles (*e.g.* the rotation and translation of the object). Such separation of modeling duties can largely alleviate the structural discontinuity problem in our model. For data synthesis, we sample $K_i$ and $z$ as an input to the generator. $K_i$ represents one of underlying clusters of training data that the DMN learns in a form of key vectors. Thus, $z$ does not need to care about the class discontinuity but focus on the attributes or styles of synthesized samples.

Our model suffers less from the forgetting problem, intuitively thanks to the explicit memory. The memory network memorizes high-level representation of clusters of real and fake samples in a form of key vectors. Moreover, the DMN allocates memory slots even for infrequent real samples while maintaining the learned slot histogram $h$ nonzero for them (*i.e.* $p(c|v_c = 1) \neq 0$). It opens a chance of sampling from rare ones for a synthesized sample, although their chance could be low.

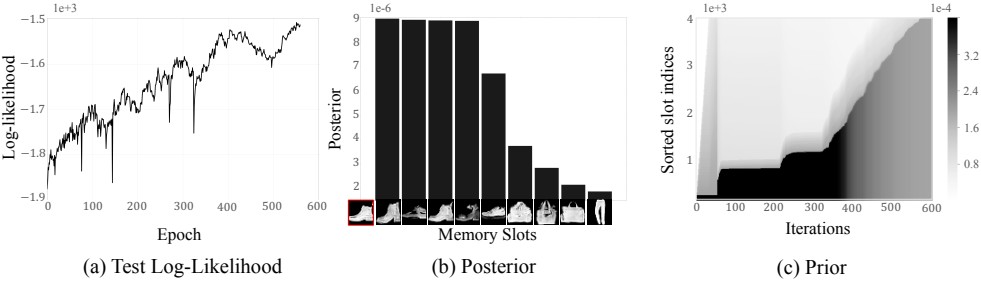

(a) Test Log-Likelihood  (b) Posterior  (c) Prior

Figure 3: Some probabilistic interpretations of MemoryGAN. (a) The biased log-likelihood $\tilde{p}(x|y = 1)$ of $10,000$ real test images (*i.e.* unseen for training) according to training epochs. (b) Examples of posteriors $p(c|x)$ of nine randomly selected memory slots for a given input image shown at the left-most column in the red box. (c) The categorical prior distribution $p(c)$ along training iterations. We sort all $N$ slots and remove the ones with too low probabilities within $[0, 0.0004]$ for readability.

## 4 EXPERIMENTS

We present some probabilistic interpretations of memoryGAN to understand how it works in section 4.1. We show both qualitative and quantitative results of image generation in section 4.2 on Fashion-MNIST (Xiao et al., 2017), CelebA (Liu et al., 2015), CIFAR10 (Krizhevsky, 2009), and Chairs (Aubry et al., 2014). We perform ablation experiments to demonstrate the usefulness of key components of our model in section 4.3. We present more experimental results in Appendix B.

For MNIST and Fashion-MNIST, we use DCGAN (Radford et al., 2016) for the inference network and generator, while for CIFAR10 and CelebA, we use the WGAN-GP ResNet (Gulrajani et al., 2017) with minor changes such as using layer normalization (Ba et al., 2016) instead of batch normalization and using ELU activation functions instead of ReLU and Leaky ReLU. We use minibatches of size $64$, a learning rate of $2 \cdot 10^{-4}$, and Adam (Kingma & Ba, 2014) optimizer for all experiments. More implementation details can be found in Appendix B.

### 4.1 PROBABILISTIC INTERPRETATION

Fig.3 presents some probabilistic interpretations of memoryGAN to gain intuitions of how it works. We train MemoryGAN using Fashion-MNIST (Xiao et al., 2017), with a hyperparameter setting of $z \in \mathbb{R}^2$, the memory size $N = 4,096$, the key dimension $M = 256$, the number of nearest neighbors $k = 128$, and $\lambda = 0.01$. Fig.3(a) shows the biased log-likelihood $\tilde{p}(x|y = 1)$ of $10,000$ real test images (*i.e.* unseen for training), while learning the model up to reaching its highest inception score. Although the inference network keeps updated during adversarial training, the vMF mixture model based memory successfully tracks its approximate distribution; as a result, the likelihood continuously increases. Fig.3(b) shows the posteriors $p(c|x)$ of nine randomly selected memory slots for a random input image (shown at the left-most column in the red box). Each image below the bar are drawn by the generator from its slot key vector $K_i$ and a random noise $z$. We observe that the memory slots are properly activated for the query, in that the memory slots with highest posteriors indeed include the information of the same class and similar styles (*e.g.* shapes or colors) to the query image. Finally, Fig.3(c) shows categorical prior distribution $p(c)$ of memory slots. At the initial state, the inference network is trained yet, so it uses only a small number of slots even for different examples. As training proceeds, the prior spreads along $N$ memory slots, which means the memory fully utilizes the memory slots to distribute training samples according to the clusters of data distribution.

### 4.2 IMAGE GENERATION PERFORMANCE

We perform both quantitative and qualitative evaluations on our model's ability to generate realistic images. Table 1 shows that our model outperforms state-of-the-art unsupervised GAN models in terms of inception scores with no additional divergence measure or auxiliary network on the CIFAR10 dataset. Unfortunately, except CIFAR10, there are no reported inception scores for other datasets, on which we do not compare quantitatively.

Table 1: Comparison of CIFAR10 inception scores between state-of-the-art unsupervised GAN models. Our memoryGAN achieves the highest score.

| Method | Score | Objective | Auxiliary net |
|---|---|---|---|
| *ALI* (Dumoulin et al., 2017) | $5.34 \pm 0.05$ | GAN | Inference net |
| *BEGAN* (Berthelot et al., 2017) | 5.62 | Energy based | Decoder net |
| *DCGAN* (Radford et al., 2016) | $6.54 \pm 0.067$ | GAN | - |
| *Improved GAN* (Salimans et al., 2016) | $6.86 \pm 0.06$ | GAN + historical averaging + minibatch discrimination | - |
| *EGAN-Ent-VI* (Dai et al., 2017) | $7.07 \pm 0.10$ | Energy based | Decoder net |
| *DFM* (Warde-Farley & Bengio, 2017) | $7.72 \pm 0.13$ | Energy based | Decoder net |
| *WGAN-GP* (Gulrajani et al., 2017) | $7.86 \pm 0.07$ | Wasserstein + gradient penalty | - |
| *Fisher GAN* (Mroueh et al., 2017) | $7.90 \pm 0.05$ | Fisher integral prob. metrics | - |
| *MemoryGAN* | $\mathbf{8.04 \pm 0.13}$ | GAN + mutual information | - |

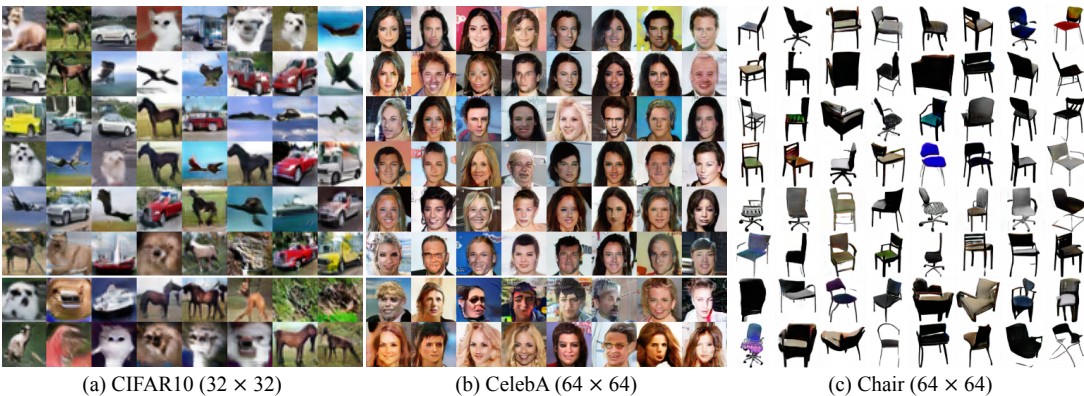

(a) CIFAR10 ($32 \times 32$)    (b) CelebA ($64 \times 64$)    (c) Chair ($64 \times 64$)

Figure 4: Image samples generated on (a) CIFAR10, (b) CelebA, and (c) Chairs dataset. Top four rows show successful examples, while the bottom two show near-miss or failure cases.

Fig.4 shows some generated samples on CIFAR10, CelebA, and Chairs dataset, on which our model achieves competitive visual fidelity. We observe that more regular the shapes of classes are, more realistic generated images are. For example, *car* and *chair* images are more realistic, while the faces of *dogs* and *cats*, *hairs*, and *sofas* are less. MemoryGAN also has failure samples as shown in 4. Since our approach is completely unsupervised, sometimes a single memory slot may include similar images from different classes, which could be one major reason of failure cases. Nevertheless, significant proportion of memory slots of MemoryGAN contain similar shaped single class, which leads better quantitative performance than existing unsupervised GAN models.

### 4.3 ABLATION STUDY

We perform a series of ablation experiments to demonstrate that key components of MemoryGAN indeed improve the performance in terms of inception scores. The variants are as follows. (i) (– EM) is our MemoryGAN but adopts the memory updating rule of Kaiser et al. (2017) instead (*i.e.* $K^t \leftarrow \frac{K^{t-1}+q}{\|K^{t-1}+q\|}$). (ii) (– MCGN) removes the slot-sampling process from the generative network. It is equivalent to DCGAN that uses the DMN as discriminator. (iii) (– Memory) is equivalent to the original DCGAN. Results in Table 2 show that each of proposed components of our MemoryGAN makes significant contribution to its outstanding image generation performance. As expected, without the memory network, the performance is the worst over all datasets.

### 4.4 GENERALIZATION EXAMPLES

We present some examples of generalization ability of MemoryGAN in Fig.5, where for each sample produced by MemoryGAN (in the left-most column), the seven nearest images in the training set are shown in the following columns. As the distance metric for nearest search, we compute the cosine similarity between normalized $q = \mu(x)$ of images $x$. Apparently, the generated images and the

Table 2: Variation of inception scores when removing one of key components of our memoryGAN on CIFAR10, affine-MNIST and Fashion-MNIST datasets.

| Variants | CIFAR10 | affine-MNIST | Fashion-MNIST |
|---|---|---|---|
| *MemoryGAN* | **8.04 ± 0.13** | **8.60 ± 0.02** | **6.39 ± 0.29** |
| (– EM) | 6.67 ± 0.17 | 8.17 ± 0.04 | 6.05 ± 0.28 |
| (– MCGN) | 6.39 ± 0.02 | 8.11 ± 0.03 | 6.02 ± 0.27 |
| (– Memory) | 5.35 ± 0.17 | 8.00 ± 0.01 | 5.82 ± 0.19 |

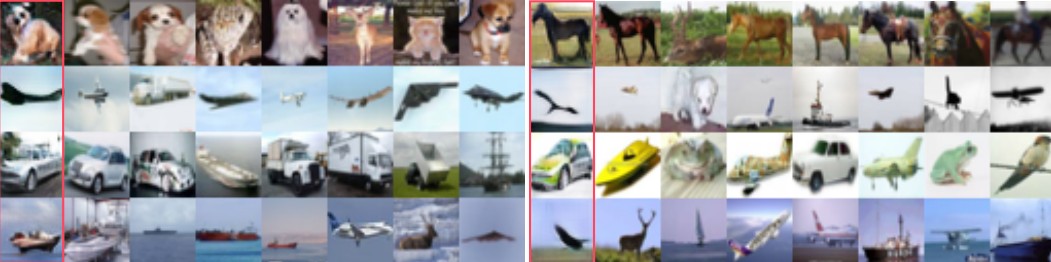

Figure 5: Examples of generalization ability of MemoryGAN. For each sample produced by MemoryGAN (in the left-most column), the seven nearest training images are shown in the following columns. The generated images and the nearest training images are quite different, which means the MemoryGAN indeed creates novel images rather than memorizing and retrieving training images.

nearest training images are quite different, which means our MemoryGAN generates novel images rather than merely memorizing and retrieving the images in the training set.

## 4.5 COMPUTATIONAL OVERHEAD

Our memoryGAN can resolve the two training problems of unsupervised GANs with mild increases of training time. For example of CIFAR10, we measure training time per epoch for MemoryGAN with 4,128K parameters and DCGAN with 2,522K parameters, which are 135 and 124 seconds, respectively. It indicates that MemoryGAN is only 8.9% slower than DCGAN for training, even with a scalable memory module. At test time, since only generator is used, there is no time difference between MemoryGAN and DCGAN.

## 5 CONCLUSION

We proposed a novel end-to-end unsupervised GAN model named *memoryGAN*, that effectively learns a highly multi-modal latent space without suffering from structural discontinuity and forgetting problems. Empirically, we showed that our model achieved the state-of-the-art inception scores among unsupervised GAN models on the CIFAR10 dataset. We also demonstrated that our model generates realistic image samples of high visual fidelity on Fashion-MNIST, CIFAR10, CelebA, and Chairs datasets. As an interesting future work, we can extend our model for few-shot generation that synthesizes rare samples.

ACKNOWLEDGEMENTS

We thank Yookoon Park, Yunseok Jang, and anonymous reviewers for their helpful comments and discussions. This work was supported by Samsung Advanced Institute of Technology, Samsung Electronics Co., Ltd, and Basic Science Research Program through the National Research Foundation of Korea (NRF) (2017R1E1A1A01077431). Gunhee Kim is the corresponding author.

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

Table 3: Summary of experimental settings.

|  | affine-MNIST | Fashion-MNIST | CIFAR10 | Chair | CelebA |
|---|---|---|---|---|---|
| Test sample size | $640,000$ | $64,000$ | $640,000$ | - | - |
| Test classifier (Acc.) | AlexNet (0.9897) | ResNet (0.9407) | Inception Net. | - | - |
| Noise dimension | 2 | 2 | 16 | 16 | 16 |
| Key dimension $M$ | 256 | 256 | 512 | 512 | 512 |
| Memory size $N$ | $4,096$ | $4,096$ | $16,384$ | $16,384$ | $16,384$ |
| Image size | $40 \times 40$ | $28 \times 28$ | $32 \times 32$ | $64 \times 64$ | $64 \times 64$ |
| Learning rate decay | None | None | Linear | Linear | Linear |

## A  APPENDIX A: MORE TECHNICAL DETAILS

### A.1  DERIVATION OF THE OBJECTIVE FUNCTION

Our objective is based on that of InfoGAN (Chen et al., 2016), which maximizes the mutual information between a subset of latent variables and observations. We add another mutual information loss term between $K_i$ and $G(z, K_i)$, to ensure that the structural information is consistent between a sampled memory key vector $K_i$ and a generated sample $G(z, K_i)$ from it:

$$
\begin{aligned}
I(K_i; G(z, K_i)) &= H(K_i) - H(K_i | G(z, K_i)) \\
&\geq H(K_i) + E_{x \sim G(z, K_i)}[E_{q \sim P(K_i|x)}[-\log Q(q|x)]] \\
&= H(K_i) + E_{x \sim G(z, K_i)}[E_{q \sim P(K_i|x)}[\kappa K_i^T q - \log C(\kappa)]]
\end{aligned}
\tag{9}
$$

where $Q$ is a von Mises-Fisher distribution $C(\kappa)exp(\kappa K_i^T q)$. Maximizing the lower bound is equivalent to minimizing the expectation term $\hat{I} = -E_{x \sim G(z, K_i)}[E_{q \sim P(K_i|x)}[\kappa K_i^T q]]$. The modified GAN objective can be written with the lower bound of mutual information, with a hyperparameter $\lambda$ (we use $\lambda = 2 \cdot 10^{-6}$).

$$
\begin{aligned}
\mathcal{L}_D &= -E_{x \sim p(x)}[\log D(x)] - E_{(z,c) \sim p(z,c)}[\log(1 - D(G(z, K_i)))] + \lambda \hat{I}. \\
\mathcal{L}_G &= E_{(z,c) \sim p(z,c)}[\log(1 - D(G(z, K_i)))] + \lambda \hat{I}.
\end{aligned}
\tag{10}
$$

## B  APPENDIX B: MORE EXPERIMENTAL RESULTS

### B.1  EXPERIMENTAL SETTINGS

We summarize experimental settings used for each dataset in Table 3. The test sample size indicates the number of samples that we use to evaluate inception scores. For Chair and CelebA dataaset, we do not compute the inception scores. The test classifier indicates which image classifier is used for computing inception scores. For affine-MNIST and Fashion-MNIST, we use a simplified AlexNet and ResNet, respectively. The noise dimension is the dimension of $z$, and the image size indicates the height and width of training images. For CIFAR10, Chair, and CelebA, we linearly decay the learning rate: $\alpha^t = \alpha^0 \cdot (1 - \gamma^{-1} \cdot t/1,000)$, where $t$ is the iteration, and $\gamma$ is the number of minibatches per epoch.

### B.2  MORE INTERPOLATION EXAMPLES FOR VISUALIZING STRUCTURE IN A LATENT SPACE

We present more interpolation examples similar to Fig.1. We use Fashion-MNIST and affine-transformed MNIST, in which we randomly rotate MNIST images by $20°$ and randomly place it in a square image of 40 pixels. First, we randomly fix a memory slot $c$, and randomly choose four noise vectors $\{z_1, \ldots, z_4\}$. Then, we interpolate the four noise vectors to extract intermediate noise vectors $\hat{z} = (1 - e)z_i + ez_j$, where $i, j \in \{1, 2, 3, 4\}$ and $e \in (0, 1)$, and visualize the generated samples $G(\hat{z}, c)$. Fig.6 shows the results. We set the memory size $N = 4,096$, the key dimension $M = 256$, the number of nearest neighbors $k = 128$, and $\lambda = 0.01$ for both datasets. We use $z \in \mathbb{R}^{32}$ for DCGAN and $z \in \mathbb{R}^2$ for MemoryGAN. In both datasets, the DMN successfully learns to partition a latent space into multiple modes, via the memory module using a von Mises-Fisher mixture model. Fig.7 presents results of the same experiments on the Chair dataset. We here show

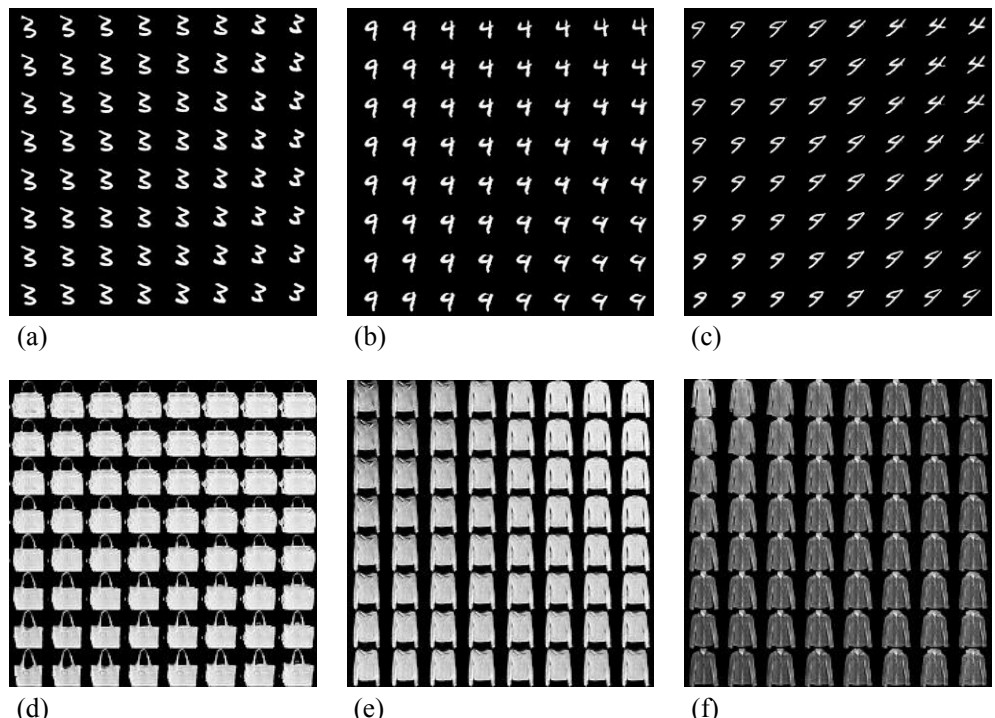

Figure 6: More interpolation examples generated by MemoryGAN. Samples are generated by fixing a discrete latent variable $c$ and sampling 64 continuous latent variable $z$. Some memory slots contain single stationary structure as shown in (a) and (d), while other slots contain images of the same class, but variation of styles as shown in (b)-(c) and (e)-(f).

examples of failure cases in (d), where interestingly some unrealistic chair images are observed as the back of chair rotates from left to right.

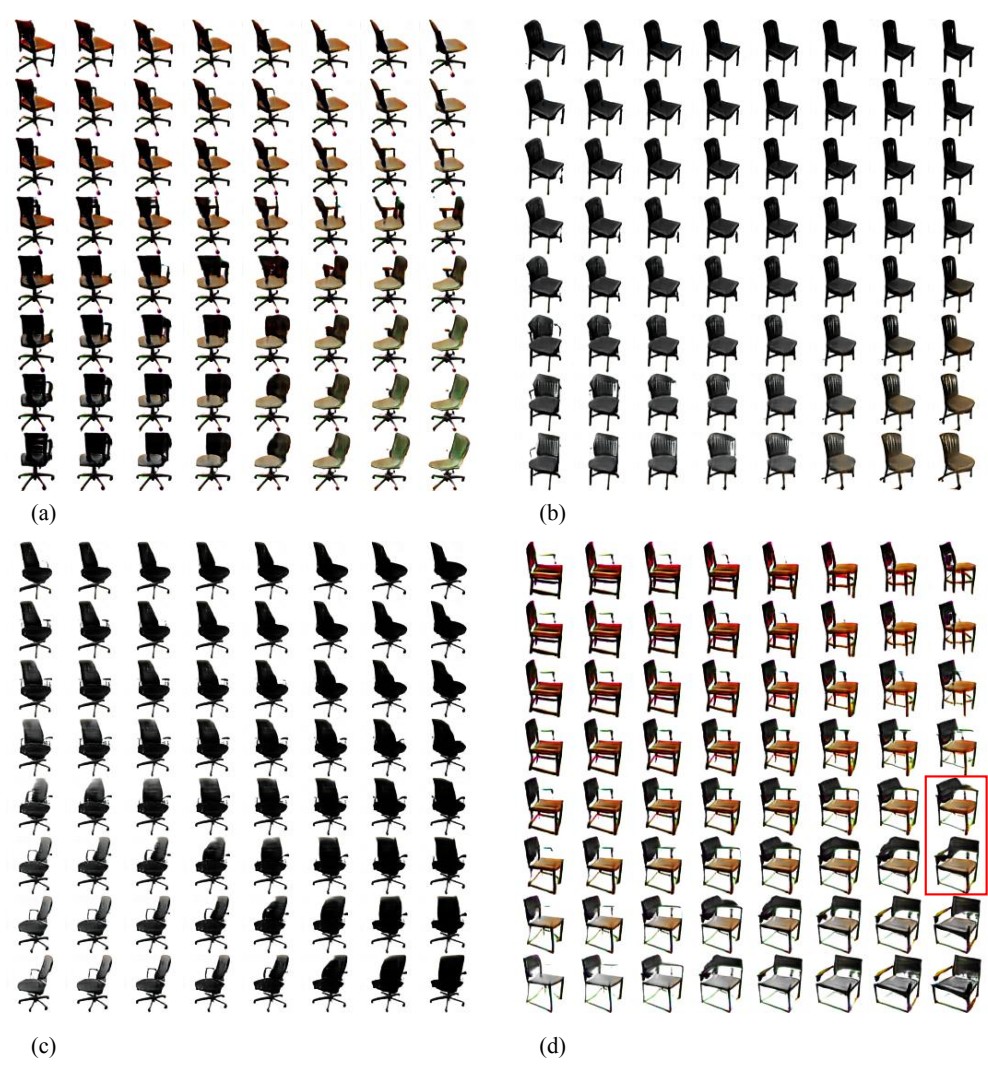

Figure 7: Interpolation examples generated by *MemoryGAN* on the Chair dataset. Memory slots contain different shapes, colors and angles of chairs as in (a)-(c). In the failure case (d), there are some unrealistic chair images as the back of chair rotates from left to right.

