# OpenReview forum: "Memorization Precedes Generation: Learning Unsupervised GANs with Memory Networks"
_ICLR.cc/2018/Conference — Accept (Poster)_

### Official Review · AnonReviewer1 · 2017-11-25
**Comments on the probabilistic interpretation, writing and the generalization ability**

**Rating:** 6
**Confidence:** 4

**Review:**

In summary, the paper introduces a memory module to the GANs to address two existing problems: (1) no discrete latent structures and (2) the forgetting problem. The memory provides extra information for both the generation and the discrimination, compared with vanilla GANs. Based on my knowledge, the idea is novel and the Inception Score results are excellent. However, there are several major comments should be addressed, detailed as follows:

1. The probabilistic interpretation seems not correct.

According to Eqn (1), the authors define the likelihood of a sample x given a slot index c as p(x|c=i) = N(q; K_i, sigma^2), where q is the normalized output of a network mu given x. It seems that this is not a well defined probability distribution because the Gaussian distribution is defined over the whole space while the support of q is restricted within a simplex due to the normalization. Then, the integral over x should be not equal to 1 and hence all of the probabilistic interpretation including the equations in the Section 3. and results in the Section 4.1. are not reliable. I'm not sure whether there is anything misunderstood because the writing of the Section 3 is not so clear.

2. The writing of the Section 3 should be improved.

Currently, the Section 3 is not easy to follow for me due to the following reasons. First, there lacks a coherent description of the notations. For instance, what's the difference between x and x', used in Section 3.1.1 and 3.1.2 respectively? According to the paper, both denote a sample. Second, the setting is somewhat unclear. For example,  it is not natural to discuss the posterior without the clear definition of the likelihood in Eqn (1). Third, a lot of details and comparison with other methods should be moved to other parts and the summary of the each part should be stated explicitly and clearly before going into details.

3. Does the large memory hurt the generalization ability of the GANs?

First of all, I notice that the random noise is much lower dimensional than the memory, e.g. 2 v.s. 256 on affine-MNIST. Does such large memory hurt the generalization ability of GANs? I suspect that most of the information are stored in the memory and only small change of the training data is allowed. I found that the samples in Figure 1 and Figure 5 are very similar and the interpolation only shows a very small local subspace near by a training data, which cannot show the generalization ability. Also note that the high Inception Score cannot show the generalization ability as well because memorizing the training data will obtain the highest score. I know it's hard to evaluate a GAN model but I think the authors can at least show the nearest neighbors in the training dataset and the training data that maximizes the activation of the corresponding memory slot together with the generated samples to see the difference.

Besides, personally speaking, Figure 1 is not so fair because a MemoryGAN only shows a very small local subspace near by a training data while the vanilla GAN shows a large subspace, making the quality of the generation different. The MemoryGAN also has failure samples in the whole latent space as shown in Figure 4.

Overall, I think this paper is interesting but currently it does not reach the acceptance threshold.

I change the rating to 6 based on the revised version, in which most of the issues are addressed.

---

> ### Author Response · Authors · 2017-12-22
> **Reply for Reviewer 1**
>
> We thank Reviewer 1 for positive and constructive reviews. Below, we respond each comment in details. Please see blue fonts in the newly uploaded draft to check how our paper is updated.
>
> 1. The probabilistic interpretation.
> Thanks for pointing out the unclearness of our formulation. First of all, the normalizing constant does not affect the model formulation, because it is a common denominator in the posterior. However, as Reviewer 1 pointed out, we use distributions on a unit sphere, and thus they should be Von Mises-Fisher (vMF) distributions with a concentration constant k=1, instead of Gaussian distributions. Without changing any fundamentals of MemoryGAN, we change the Gaussian mixtures to Von Mises-Fisher Mixtures in the draft. We appreciate Reviewer 1 for the correction.
>
> 2. Writing improvement of Section 3.
> (1) The difference between x and x' in Section 3.1.1-3.1.2.
> We used x to denote samples for updating discriminator parameters, and x’ for updating the memory module. Since every training sample goes through these two updating operations, there is no need to use both, and we unify them to x.
> (2) Discuss the posterior without the clear definition of the likelihood in Eq.(1).
> The likelihood for Eq.(1) is identical to that of the standard vMF mixture model. Thus, we omitted it and directly introduced the posterior equation. We will clarify them.
> (3) Overall organization
> We will re-organize the draft so that key ideas in each part are explicitly summarized before the details.
>
> 3. Generalization ability.
> As Reviewer 1 suggested, we add an additional result to the Figure 5, where for each sample produced by MemoryGAN (in the left-most column), the seven nearest images in the training set are shown in the following columns. Apparently, our MemoryGAN generates novel images rather than merely memorizing and retrieving the images in the training set.
> The memory is used to represent not only positive samples but also possible fake samples. Thus, the memory size is rather large (n=16384), for the CIFAR10 dataset. That is, the more diverse the dataset is, the larger memory size is required to represent both variability. In our experiments, we set the memory size based on the performance on the validation set.
>
> 4. Fig.1.
> Initially, we fixed the discrete latent variable c for MemoryGAN, because it is a memory index, and thus it is meaningless to interpolate over c. However, we follow Reviewer’s rationale and update Fig.1 in the new draft. Please check it.
> In new Fig.1.(b,d), we first randomly sample both (z,c) shown at the four corners in blue boxes. We then generate 64 images by interpolating both (z,c). However, since the interpolation over c is meaningless, we take key values K_c of the four randomly sampled c’s, and then perform interpolation over their K_c’s. Then, for each interpolated K_c’, we find the memory slot c = argmax p(c|K_c’), i.e. the memory index whose posterior is the highest with respect to K_c’.
> As shown in Fig.1.(b,d), different classes are shown at the four corners, and other samples gradually change, but no structural discontinuity occurs. We hope the modified Fig.1 delivers the merits of MemoryGAN more intuitively.
>
> 5. Failure cases of Fig.4.
> As Reviewer 1 pointed out, MemoryGAN also has failure samples in the whole latent space as shown in Figure 4. Since our approach is completely unsupervised, sometimes a single memory slot may include similar images from different classes. It causes failure cases. Nevertheless, significant proportion of memory slots of MemoryGAN contain similar shaped single class, which leads much better performance than existing unsupervised GAN models.

---

> > ### Comment · AnonReviewer1 · 2018-01-01
> > **Change rating to 6**
> >
> > Thanks for the detailed rebuttal. I'm glad to see most of the issues are addressed in the revision and I'd like to change the rating to 6.

---

> > > ### Author Response · Authors · 2018-01-02
> > > **We'd like to appreciate Reviewer 1**
> > >
> > > We'd like to appreciate Reviewer 1 again for the constructive comments, which are greatly helpful to make our paper better. We are very glad that our rebuttal clarifies Reviewer 1's concerns.

---

### Official Review · AnonReviewer2 · 2017-11-27
**An interesting idea with clear demonstration**

**Rating:** 6
**Confidence:** 4

**Review:**

MemoryGAN is proposed to handle structural discontinuity (avoid unrealistic samples) for the generator, and the forgetting behavior of the discriminator. The idea to incorporate memory mechanism into GAN is interesting, and the authors make nice interpretation why this needed, and clearly demonstrate which component helps (including the connections to previous methods).

My major concerns:

Figure 1 is questionable in demonstrating the advantage of proposed MemoryGAN. My understanding is that four z's used in DCGAN and MemoryGAN are "randomly sampled" and fixed, interpolation is done in latent space, and propagate to x to show the samples.  Take MNIST for example, It can be seen that the DCGAN has to (1) transit among digits in different classes, while MemoryGAN only (2) transit among digits in the same class. Task 1 is significantly harder than task 2, it is not surprise that DCGAN generate unrealistic images. A better experiment is to fix four digits from different class at first, find their corresponding latent codes, do interpolation, and propagate back to sample space to visualize results. If the proposed technique can truly handle structural discontinuity, it will "jump" over the sample manifold from one class to another, and thus avoid unrealistic samples. Also, the current illustration also indicates that the generated samples by MemoryGAN is not diverse.

It seems the memory mechanism can bring major computational overhead, is it possible to provide the comparison on running time?

To what degree the MemoryGAN can handle structural discontinuity? It can be seen from Table 2 that larger improvement is observed when tested on a more diverse dataset. For example, the improvement gap from MNIST to CIFAR is larger. If the MemoryGAN can truly deal with structural discontinuity, the results on generating a wide range of different images for ImageNet may endow the paper with higher impact.

The authors should consider to make their code reproducible and public.


Minor comments:

In Section 4.3, Please fix "Results in 2" as "Results in Table 2".

---

> ### Author Response · Authors · 2017-12-22
> **Reply for Reviewer 2**
>
> We thank Reviewer 2 for positive and constructive reviews. Below, we respond each comment in details. Please see blue fonts in the newly uploaded draft to check how our paper is updated.
>
> 1. Fig.1.
> Initially, we fixed the discrete latent variable c for MemoryGAN, because it is a memory index, and thus it is meaningless to interpolate over c. However, we follow Reviewer’s rationale and update Fig.1 in the new draft. Please check it.
> In new Fig.1.(b,d), we first randomly sample both (z,c) shown at the four corners in blue boxes. We then generate 64 images by interpolating both (z,c). However, since the interpolation over c is meaningless, we take key values K_c of the four randomly sampled c’s, and then perform interpolation over their K_c’s. Then, for each interpolated K_c’, we find the memory slot c = argmax p(c|K_c’), i.e. the memory index whose posterior is the highest with respect to K_c’.
> As shown in Fig.1.(b,d), different classes are shown at the four corners, and other samples gradually change, but no structural discontinuity occurs. We hope the modified Fig.1 delivers the merits of MemoryGAN more intuitively.
>
> 2. Computation overhead.
> As we replied to Reviewer 2, we measure the training time per epoch for MemoryGAN (4,128K parameters) and DCGAN (2,522K parameters), which are 135 sec and 124 sec, respectively.
> It means MemoryGAN is only 8.9% slower than DCGAN for training, even with a scalable memory module. At test time, since only generator is used, there is no time difference between MemoryGAN and DCGAN.
>
> 3. ImageNet experiments.
> We observed that the memory module significantly helps improve the performance when using highly diverse datasets. For example, inception scores are higher for CIFAR10 than for FashionMNIST. Thus, as Reviewer 2 suggested, we can easily expect that the our MemoryGAN works better for the ImageNet dataset. We did not test with ImageNet, mainly because of too long training time (more than two weeks by our estimation). However, we will do it as a future work.
>
> 4. Source code and typos.
> We plan to make public the source code.
> Thank you for correct typos!

---

### Official Review · AnonReviewer3 · 2017-11-27
**Review from AnonReviewer3**

**Rating:** 7
**Confidence:** 4

**Review:**

[Overview]

In this paper, the authors proposed a novel model called MemoryGAN, which integrates memory network with GAN. As claimed by the authors, MemoryGAN is aimed at addressing two problems of GAN training: 1) difficult to model the structural discontinuity between disparate classes in the latent space; 2) catastrophic forgetting problem during the training of discriminator about the past synthesized samples by the generator. It exploits the life-long memory network and adapts it to GAN. It consists of two parts, discriminative memory network (DMN) and Memory Conditional Generative Network (MCGN). DMN is used for discriminating input samples by integrating the memory learnt in the memory network, and MCGN is used for generating images based on random vector and the sampled memory from the memory network. In the experiments, the authors evaluated memoryGAN on three datasets, CIFAR-10, affine-MNIST and Fashion-MNIST, and demonstrated the superiority to previous models. Through ablation study, the authors further showed the effects of separate components in memoryGAN.

[Strengths]

1. This paper is well-written. All modules in the proposed model and the experiments were explained clearly. I enjoyed much to read the paper.

2. The paper presents a novel method called MemoryGAN for GAN training. To address the two infamous problems mentioned in the paper, the authors proposed to integrate a memory network into GAN. Through memory network, MemoryGAN can explicitly learn the data distribution of real images and fake images. I think this is a very promising and meaningful extension to the original GAN.

3. With MemoryGAN, the authors achieved best Inception Score on CIFAR-10. By ablation study, the authors demonstrated each part of the model helps to improve the final performance.

[Comments]

My comments are mainly about the experiment part:

1. In Table 2, the authors show the Inception Score of images generated by DCGAN at the last row. On CIFAR-10, it is ~5.35. As the authors mentioned, removing EM, MCGCN and Memory will result in a conventional DCGAN. However, as far as I know, DCGAN could achieve > 6.5 Inception Score in general.  I am wondering what makes such a big difference between the reported numbers in this paper and other papers?

2. In the experiments, the authors set N = 16,384, and M = 512, and z is with dimension 16. I did not understand why the memory size is such large. Take CIFAR-10 as the example, its training set contains 50k images. Using such a large memory size, each memory slot will merely count for several samples. Is a large memory size necessary to make MemoryGAN work? If not, the authors should also show ablated study on the effect of different memory size; If it is true, please explain why is that. Also, the authors should mention the training time compared with DCGAN. Updating memory with such a large size seems very time-consuming.

3. Still on the memory size in this model. I am curious about the results if the size is decreased to the same or comparable number of image categories in the training set. As the author claimed, if the memory network could learn to cluster training data into different category, we should be able to see some interesting results by sampling the keys and generate categoric images.

4. The paper should be compared with InfoGAN (Chen et al. 2016), and the authors should explain the differences between two models in the related work. Similar to MemoryGAN, InfoGAN also did not need any data annotations, but could learn the latent code flexibly.

[Summary]

This paper proposed a new model called MemoryGAN for image generation. It combined memory network with GAN, and achieved state-of-art performance on CIFAR-10. The arguments that MemoryGAN could solve the two infamous problem make sense. As I mentioned above, I did not understand why the authors used such large memory size. More explanations and experiments  should be conducted to justify this setting. Overall, I think MemoryGAN opened a new direction of GAN and worth to further explore.

---

> ### Author Response · Authors · 2017-12-22
> **Reply for Reviewer 3**
>
> We thank Reviewer 3 for positive and constructive reviews. Below, we respond each comment in details. Please see blue fonts in the newly uploaded draft to check how our paper is updated.
>
> 1. DCGAN inception scores.
> Thanks for a correction. As R3 pointed out, the DCGAN inception score of the original paper is 6.54+-0.67. The value 5.35 that we reported previously was the score of “MemoryGAN without memory”, which is identical to the DCGAN in terms of model structure. That was the reason why we named it as DCGAN. However, the “MemoryGAN without memory” had different details from the DCGAN, including the ELU activation (instead of ReLU and Leaky ReLU) and layer-normalization (instead of batch normalization). To resolve the confusion, we change the values of Table 1 to 6.54+-0.67 (the numbers reported in the original DCGAN paper).
>
> 2. The memory size of MemoryGAN.
> In our experiments, we set the memory size based on the performance on the validation set. The memory is used to represent not only positive samples but also possible fake samples. Thus, the memory size is rather large (n=16384), for the CIFAR10 dataset whose size is 50,000. That is, the more diverse the dataset is, the larger memory size is required to represent both variability. When we used a half-size memory (n=8192), the inception score for CIFAR10 decreased from 8.04 to 6.71.
> As Reviewer 3 suggested, we test with decreasing the memory size to n=16, which is similar to the number of classes, on the Fashion-MNIST and CIFAR10 datasets. We obtain the inception score 6.14 for Fashion-MNIST with n=16, which is slightly lower than the reported score 6.39 with n=4096. On the other hand, for CIFAR10, the inception score significantly decreases from 8.04 with n=16384 to 3.06 with n=16. These results indicate that intra-class variability of Fashion-MNIST is small, while that of CIFAR10 is very high.
>
> 3. Training/Test time.
> The training time per epoch for MemoryGAN (4,128K parameters) and DCGAN (2,522K parameters) are 135 sec and 124 sec, respectively. It means MemoryGAN is only 8.9% slower than DCGAN for training, even with a scalable memory module. At test time, since only the generator is used, there is no time difference between MemoryGAN and DCGAN.
>
> 4. Comparison with InfoGAN.
> There are two key differences between InfoGAN and MemoryGAN. First, InfoGAN implicitly learns the latent cluster information of data into model parameters, while MemoryGAN explicitly maintains the information about the whole training set using a life-long memory network. Thus, MemoryGAN keeps track of current cluster information stably and flexibly without suffering from forgetting old samples. Second, MemoryGAN explicitly offers various distributions like prior distribution p(c), conditional likelihood p(x|c) and marginal likelihood p(x), unlike InfoGAN. Such interpretability is useful for designing or training the models.

---

### Decision · Program_Chairs · 2018-01-29
**ICLR 2018 Conference Acceptance Decision**

**Decision:**

Accept (Poster)

**Comment:**


I am going to recommend acceptance of this paper despite being worried about the issues raised by reviewer 1.  In particular,

1:  the best possible inception score would be obtained by copying the training dataset
2:  the highest visual quality samples would be obtained by copying the training dataset
3:  perturbations (in the hidden space of a convnet) of training data might not be perturbations in l2, and so one might not find a close nearest neighbor with an l2 search
4:  it has been demonstrated in other works that perturbations of convnet features of training data (e.g. trained as auto-encoders) can make convincing "new samples"; or more generally, paths between nearby samples in the hidden space of a convnet can be convincing new samples.

These together suggest the possibility that the method presented is not necessarily doing a great job as a generative model or as a density model (it may be, we just can't tell...), but it is doing a good job at hacking the metrics (inception score, visual quality).      This is not an issue with only this paper, and I do not want to punish the authors of this papers for the failings of the field; but this work, especially because of its explicit use of training examples in the memory,  nicely exposes the deficiencies in our community's methodology for evaluating GANs and other generative models.